# The Mechanism of Steroid Hormones in Non-Small Cell Lung Cancer: From Molecular Signaling to Clinical Application

**DOI:** 10.3390/biomedicines13081992

**Published:** 2025-08-15

**Authors:** Yao Wang, Ying Zhou, Yao Yao, Caihong Zheng

**Affiliations:** 1Department of Pharmacy, Women’s Hospital School of Medicine, Zhejiang University, Hangzhou 310006, China; jidantou@outlook.com; 2Department of Clinical Pharmacology, Affiliated Hangzhou First People’s Hospital, School Of Medicine, Westlake University, Hangzhou 310006, China; zypharm@zju.edu.cn

**Keywords:** steroid hormones, non-small cell lung cancer, hormone receptors, glucocorticoids, targeted therapy

## Abstract

Steroid hormones play critical roles in the development and progression of NSCLC through both genomic and non-genomic pathways. This review summarizes the expression profiles and molecular functions of estrogen, progesterone, androgen, and glucocorticoid receptors in NSCLC. Estrogen and progesterone receptors exhibit gender-specific prognostic significance, while glucocorticoid receptors influence tumor growth and immune responses. Emerging evidence supports the use of anti-estrogen therapies and glucocorticoids as adjuncts to existing treatment strategies, including immunotherapy. The crosstalk between hormone signaling and oncogenic pathways such as EGFR or immune checkpoints offers opportunities for novel combination therapies. However, challenges remain in biomarker development, drug resistance, and managing the dual effects of glucocorticoids. A deeper understanding of hormone–tumor–immune interactions is essential to optimize hormone-targeted interventions in NSCLC.

## 1. Introduction

### 1.1. An Epidemiological Overview of Non-Small Cell Lung Cancer

Non-small cell lung cancer (NSCLC) represents the most prevalent form of lung cancer and is among the most lethal malignancies globally. The incidence of NSCLC rises with advancing age [1]. According to global statistics from 2022, approximately 20 million new cancer cases were reported worldwide, resulting in nearly 10 million deaths. The World Health Organization (WHO) projects that the global cancer burden will increase by approximately 60% over the next two decades [2]. Population-based projections indicate that, by the year 2050, the annual number of new cancer cases may surpass 35 million, reflecting a 77% increase compared to 2022 figures. In 2022, lung cancer remained the leading cause of cancer-related morbidity and mortality worldwide, with nearly 2.5 million new cases diagnosed and more than 1.8 million deaths reported. It accounted for approximately one-eighth (12.4%) of all newly diagnosed cancers and nearly one-fifth (18.7%) of all cancer-related deaths globally [3].

### 1.2. Basic Concepts and Classification of Steroid Hormones

Steroid hormones, a class of bioactive compounds derived from cholesterol metabolism, are synthesized through a complex enzymatic pathway. These hormones are predominantly produced in specific endocrine tissues: glucocorticoids are primarily synthesized in the adrenal cortex, androgens in the testes, and estrogens in the ovaries and placenta. Although the minor extraglandular synthesis of these hormones may occur, their overall biosynthesis is tightly regulated by intricate physiological mechanisms. For instance, glucocorticoid production is influenced by circadian rhythms and stress responses and is primarily controlled through the hypothalamic–pituitary–adrenal axis. In contrast, the synthesis of androgens and estrogens is predominantly regulated by the hypothalamic–pituitary–gonadal axis, which also plays a pivotal role in the regulation of the menstrual cycle [4]. As small lipophilic molecules, steroids possess the ability to readily cross cell membranes and the blood–brain barrier, enabling widespread distribution and action throughout the body. Once in circulation, these hormones can reach virtually all tissues and cells. Traditionally, the spatial specificity of steroid hormone action has been considered to be determined by the presence and expression levels of specific steroid receptors in target cells. However, emerging evidence indicates that active steroid hormones can also be synthesized locally within target tissues or generated de novo from inactive circulating precursors, thereby enabling the localized regulation of steroid signaling. Steroids that are synthesized and exert functional effects within the central nervous system are specifically referred to as neurosteroids. Classical neurosteroids include four major categories: progesterone, estrogen, androgens, and corticosteroids. These molecules not only function as systemic endocrine signals but also modulate neuronal activity in a paracrine manner, contributing to the fine-tuned regulation of local neural processes [5].

### 1.3. The Association Between Steroid Hormones and Cancer

Against the backdrop of the rising global incidence and mortality of lung cancer, steroid hormones—important bioactive molecules in human physiology—have increasingly become a focus of research in the pathogenesis, progression, and therapeutic strategies of NSCLC. Accumulating evidence indicates that one of the most prominent biological distinctions between males and females is the differential expression of sex hormones, which may play a crucial role in the mechanisms underlying lung cancer development and progression. Epidemiological data suggest that women are more susceptible to the carcinogenic components present in tobacco smoke than men [6]. Under comparable exposure conditions, particularly in recent years, the incidence of lung cancer among women has risen significantly and exceeds that observed in men. Moreover, even among never-smokers, the annual incidence of lung cancer in women continues to increase and remains higher than in their male counterparts [7]. Hormonal status is rarely considered in studies; however, when premenopausal women are analyzed independently, they are more frequently diagnosed at advanced stages, exhibit lower tumor differentiation, have a higher incidence of metastasis, and experience a worse prognosis compared to postmenopausal women and men [8]. Recently, it has been reported (n = 1104) that the overall survival rate among premenopausal women is lower than that of men and postmenopausal women, which supports the involvement of sex hormones in the progression of lung cancer [9]. These findings collectively highlight the significant involvement of sex steroid hormones—particularly estrogens and progestogens—in the biological behavior and clinical manifestations of NSCLC.

In terms of treatment modalities, lung cancer management primarily includes surgical resection, chemotherapy, radiotherapy, targeted therapy, and immunotherapy. However, both chemotherapy and radiotherapy can cause varying degrees of pulmonary tissue damage, while immunotherapy may lead to immune-related adverse events due to its non-specific activation of the immune system [10]. In the context of managing symptoms associated with tumor progression or treatment-related side effects, glucocorticoids (GCs) represent an effective therapeutic option. Nevertheless, their immunosuppressive properties not only exacerbate the immune dysfunction induced by tumor progression itself but may also compromise the efficacy of immunotherapeutic interventions [11]. Therefore, this article focuses on the application of steroid hormones in NSCLC, reviewing the roles of sex hormones and GCs. While sex hormones influence tumorigenesis and progression through gender-specific biological mechanisms, GCs interact complexly with immune function and the tumor microenvironment during clinical treatment. This review aims to provide insights into the underlying mechanisms and potential clinical applications of steroid hormones in lung cancer.

The selection criteria for this review encompassed the most recent research literature published since the year 2010. Scientific papers published prior to 2010 were also included if they demonstrated conceptual clarity and relevance. A systematic literature search was conducted using selected keywords from the PubMed database, including “lung cancer”, “steroids”, “estrogen”, “androgen”, “progestin”, and “glucocorticoid”. These keywords were applied either individually or in combination. The most recent review articles within each thematic area were selected as the primary reference sources, while articles containing duplicate or highly overlapping information were excluded.

## 2. The Expression of Steroid Hormone Receptors in NSCLC

The biological effects of steroid hormones are largely mediated through their respective receptors, including estrogen receptors (ERs), the progesterone receptor (PR), the androgen receptor (AR), and the glucocorticoid receptor (GR) (Table 1). Understanding the expression patterns of these receptors in lung cancer is crucial in elucidating their functional roles and therapeutic relevance. While traditionally underexplored in thoracic malignancies, increasing evidence indicates that these receptors are variably expressed in NSCLC, with potential implications for tumor progression, prognosis, and responsiveness to hormone-based therapies. This section provides a comprehensive overview of the expression profiles of steroid hormone receptors in lung cancer subtypes, highlighting their associations with clinicopathological features, sex-related differences, and molecular subtypes.

### 2.1. Estrogen Receptor Expression

Since Chaudhuri et al. [17] first documented the presence of ERs in lung cancer tissues in 1982, extensive research has been conducted to investigate ER expression in this context. In the field of lung cancer research, the expression of ERs has garnered significant attention due to its potential involvement in multiple aspects of tumor biology. Numerous studies have confirmed the presence of classical ERs in both normal lung tissues and lung malignancies, particularly lung adenocarcinoma [18]. However, there remains no consensus among international and domestic scholars regarding ER expression in NSCLC [19]. Some researchers argue that ERs are specifically expressed in NSCLC tissues and absent in normal or benign lung tissues, while others suggest that ERs are present across all tissue types—normal, benign, and malignant—with the highest levels observed in NSCLC. Furthermore, conflicting views exist concerning the biological role of ERs in NSCLC. Some studies indicate that patients with ER-positive tumors exhibit better clinical outcomes compared to those who are ER-negative, suggesting a potential inhibitory effect of estrogen via the ER signaling pathway [20]. Conversely, other investigations report that estrogen may promote the proliferation of NSCLC cells, implying a tumor-promoting role [21].

In lung cancer tissues, two primary isoforms of the ER have been identified: ERα and ERβ. ERα is mainly found in tissues such as the breast, ovary, and endometrium, while ERβ has a wider distribution, including lung tissues, etc. [12,13] (Table 1). The expression pattern of ERα remains controversial. Several studies have reported low or even undetectable levels of ERα in both lung cancer tissues and cell lines. Moreover, its prognostic significance remains unclear. While some reports associate ERα expression with poor survival outcomes, others find no significant correlation with patient prognosis [22]. In contrast, accumulating evidence suggests that ERβ plays a more prominent role in NSCLC. Its expression has generally been linked to improved clinical outcomes [23]. For instance, Smida et al. [24] demonstrated that both ERα and ERβ are expressed in both genders, although ERα levels are typically lower in lung cancer cells. Female NSCLC patients with ERα-positive tumors were found to have significantly better survival rates than their male counterparts, indicating that ERα may serve as an independent prognostic factor in advanced NSCLC. However, other studies have suggested that ERα expression combined with the absence of ERβ expression correlates with poorer survival outcomes [15]. Luo Z et al. [25] reported a significant association between ERβ expression and improved overall survival in NSCLC patients. In contrast, Skjefstad et al. [26] observed that high ERβ expression was associated with worse prognosis. A recent study has demonstrated that, among 4874 lung cancer cases, ERβ was the predominant receptor subtype, exhibiting an expression rate of 56.5%. In comparison, the expression rate of ERα was 33.1% [27]. Wu et al. [28] used immunohistochemical techniques to analyze 301 NSCLC tissue samples and found no detectable ERα expression, whereas ERβ was positively expressed in 45.8% of cases. This expression was found to be gender-related, with significantly higher ERβ positivity in female patients compared to males (*p* < 0.05), and was also positively correlated with the degree of tumor differentiation. The expression of ERs in lung cancer is complex and multifaceted. Variations in receptor subtype expression, their associations with prognosis, and gender-specific differences offer novel insights into lung cancer research and therapeutic strategies. Further in-depth investigation is warranted to clarify these mechanisms and their clinical implications.

### 2.2. Progesterone Receptor Expression

Progesterone serves as a key regulatory hormone in the female reproductive system, primarily secreted by luteal cells of the ovary, with progesterone itself being the principal biologically active form [29]. The cellular actions of progesterone are mediated through the PR, a member of the steroid hormone receptor superfamily and a ligand-activated transcription factor. The PR shares structural and functional similarities with the ER and is predominantly expressed in steroid-responsive tissues such as the breast, uterus, and ovary, although it can also be detected in other tissue types [29]. The PR exists in two isoforms, PR-A and PR-B, which demonstrate distinct regulatory roles in cancer progression. Nevertheless, there remains controversy regarding the prognostic significance of PR expression in NSCLC tumor tissues. While some studies suggest a positive correlation between PR expression and prolonged patient survival [30], others have failed to identify a significant association between PR levels and overall survival [15]. Romanov I.P. et al. [31] conducted a comparative analysis of survival rates among patients with NSCLC stratified by the PR expression levels in tumor tissues. The study included a total of 130 patients, in whom PR expression had been previously assessed. The results indicated that PR was detectable in all analyzed tumors, with a median expression level of 57%, whereas other studies report minimal or absent PR expression in NSCLC tissues [14] (Table 1). Furthermore, some prognostic studies suggest that low PR expression is associated with poorer clinical outcomes in NSCLC patients. Similar to the autocrine mechanisms observed for estrogen, lung cancer cells have been shown to synthesize progesterone endogenously. In NSCLC patients, PR expression does not appear to be significantly influenced by factors such as age, menopausal status, ER status, or P53 protein expression levels. Notably, higher PR positivity has been observed in female patients, early-stage tumors, and poorly differentiated NSCLC subtypes. Multivariate analyses have further confirmed that PR expression is an independent prognostic factor in NSCLC, with higher expression correlating with improved overall survival [32]. A study investigated the expression of the PR in 130 surgically resected NSCLC samples using immunofluorescence combined with flow cytometry. Following statistical analysis and the exclusion of censored events, a cutoff value of 57% was applied. The median survival time was 12.8 months for patients with high PR expression (≥57%) and 25.8 months for those with low expression (<57%). High PR expression was significantly associated with a poorer prognosis (*p* = 0.05, HR = 1.7). Therefore, the study concluded that high PR expression in NSCLC correlates with shorter patient survival, and PR modulators may hold therapeutic potential for patients with PR-positive NSCLC [31]. In contrast, findings from Raso et al. [33] indicated no significant association between PR expression and overall survival. Their data revealed a higher PR positivity rate in SCCs (70%) compared to adenocarcinomas (58%). Collectively, these findings highlight the heterogeneous nature of PR expression and function in lung cancer, underscoring the need for further investigation into its underlying mechanisms and potential clinical implications.

### 2.3. Androgen Receptor Expression

Although extensive research has demonstrated the involvement of sex hormones in the pathogenesis of lung cancer, particularly among non-smoking women, much of the existing literature has primarily focused on estrogen and progesterone. Nevertheless, accumulating evidence suggests that androgens—traditionally regarded as male sex hormones—also play a role in lung cancer development and progression [34]. The AR, which mediates the biological effects of androgens, is expressed not only in reproductive tissues but also in various non-reproductive organs under both physiological and pathological conditions. In addition to its classical target tissues, AR expression has been detected in normal human lung tissue, as well as in both NSCLC and small cell lung cancer (SCLC) [35]. Research investigating the relationship between AR and tumorigenesis is relatively recent, and studies specifically examining AR expression in NSCLC remain limited [36]. Moreover, the reported AR expression levels in lung cancer tissues exhibit considerable variability across different studies [15] (Table 1). Rades et al. [15] reported that, among 64 patients with stage II–III NSCLC who underwent radiotherapy, 31% exhibited AR expression in their tumor tissues; however, AR status was not identified as an independent prognostic factor. Interestingly, clinical observations have revealed changes in androgen levels during targeted therapy for advanced or recurrent NSCLC. Specifically, androgen levels were found to decrease significantly following treatment with gefitinib, an epidermal growth factor receptor (EGFR) tyrosine kinase inhibitor (TKI). In female patients, lower baseline androgen levels were associated with a more favorable response to gefitinib therapy.

### 2.4. Glucocorticoid Receptor Expression

GCs are a class of steroid hormones that play essential roles in maintaining physiological homeostasis. Their biological effects are primarily mediated through the GR, an intracellular, ligand-activated transcription factor that belongs to the nuclear receptor superfamily and is widely expressed across various tissues. Clinical evidence indicates that GR expression is downregulated in patients with NSCLC. Moreover, elevated GR expression has been significantly correlated with improved overall survival in this patient population [37] Using immunohistochemical analysis, Lu et al. demonstrated that approximately 50% of patients with advanced-stage NSCLC exhibit high GR expression [38]. In another study, a Kaplan–Meier survival analysis was conducted using data from 1529 lung cancer patients, the majority of whom did not undergo adjuvant therapy either pre- or postoperatively [16]. Patients were stratified into three groups based on GR activity inferred from transcriptomic profiling. The analysis demonstrated that individuals with higher GR activity had significantly improved outcomes in terms of both overall survival and recurrence-free survival when compared to those with moderate or low GR activity levels.

Dexamethasone, a synthetic glucocorticoid, activates GR signaling, which subsequently induces the expression of CDKN1C/p57, leading to G1/S-phase cell cycle arrest, specifically in Liver Kinase B1 (LKB1)-mutated NSCLC cells, thereby suppressing tumor growth. Furthermore, high expression of carbamoyl-phosphate synthetase 1 has been strongly associated with LKB1 mutation status and may serve as a potential biomarker for the prediction of sensitivity to dexamethasone treatment [39].

## 3. Molecular Signaling Pathways of Steroid Hormones in NSCLC

The biological effects of steroid hormones in NSCLC are mediated through diverse molecular signaling pathways, encompassing both genomic and non-genomic mechanisms. Upon binding to their respective receptors, steroid hormones can act as transcriptional regulators within the nucleus or trigger rapid cytoplasmic signaling cascades that modulate cell proliferation, apoptosis, angiogenesis, and immune evasion. These pathways often intersect with key oncogenic signals such as EGFR, MAPK, PI3K/AKT, and STAT3, contributing to tumor progression and therapeutic resistance. This section delineates the canonical and non-canonical signaling routes of estrogen and the GR in NSCLC, offering mechanistic insights into their roles in lung tumor biology and treatment responses.

### 3.1. Estrogen-Mediated Signaling Pathways

In the context of lung cancer, estrogen exerts its biological effects primarily through two distinct signaling pathways following its binding to the ER (Figure 1). The first pathway is genomic, wherein the estrogen–ER complex translocates to the nucleus and modulates gene transcription. The second pathway is non-genomic, in which the complex interacts with membrane-associated signaling components to regulate the activation of protein kinases, second messengers, and ion channels, ultimately promoting tumor cell proliferation in patients with NSCLC [21]. Additionally, estrogen signaling contributes to tumor angiogenesis by inducing the secretion of vascular endothelial growth factor (VEGF), thereby enhancing the proliferation of vascular endothelial cells [40,41]. Emerging evidence also indicates that estrogen can promote NSCLC progression via the G-protein-coupled estrogen receptor (GPER). GPER signaling has been detected in both the cytoplasm and nuclei of tumor cells in NSCLC specimens, and elevated expression levels of this receptor have been associated with advanced tumor stages, lower differentiation grades, and high ERβ expression. Studies have demonstrated that G15, a selective GPER inhibitor, can effectively reverse estradiol-induced cell proliferation. These findings suggest that targeting the GPER signaling pathway may represent a promising therapeutic strategy for future NSCLC treatment [42].

The major ER-mediated intracellular signaling cascades include the mitogen-activated protein kinase/extracellular signal-regulated kinase (MAPK/ERK) pathway and the c-Jun N-terminal kinase (JNK) pathway, as well as the phosphoinositide 3-kinase/protein kinase B (PI3K/Akt) and cyclic adenosine monophosphate/protein kinase A (cAMP/PKA) pathways. Moreover, estrogen can contribute to tumor development through ER-dependent mechanisms involving the formation of genotoxic metabolites such as 4-hydroxyestrogen1, 4-hydroxyestrone2, and estrogen quinone derivatives [43,44]. In lung malignancies, estrogen rapidly activates cellular responses involving MAPK and AKT, leading to the phosphorylation of steroid receptor coactivators. These molecular events are closely linked to subsequent processes such as NSCLC cell proliferation, angiogenesis, and overall tumor progression [45,46].

The small GTPases of the RAS family, including N-Ras, H-Ras, and K-Ras, are highly susceptible to mutations, leading to their constitutive activation in cancer. These proteins are closely linked to the activation of cell surface receptors. Research has shown that estrogen can activate human RAF1 in SCLC cells, thereby inhibiting cell proliferation through cell cycle arrest in the G1 and G2 phases. Furthermore, studies have indicated that RAF1 expression is associated with smoking status and gender (particularly in female patients) in NSCLC. Therefore, the RAF/MEK/MAPK signaling pathway may represent a promising therapeutic target for SCLC and other neuroendocrine tumors [47].

### 3.2. Glucocorticoid-Mediated Signaling Pathway

GCs primarily exert their biological effects through genomic mechanisms. Due to their high lipophilicity, these hormones can rapidly diffuse across cell membranes and enter the cytoplasm, where they bind to the GR, a member of the nuclear receptor superfamily [48]. The GRα isoform is the predominant functional form and is expressed in nearly all nucleated cells throughout the body [49]. Upon ligand binding, the glucocorticoid–GR complex translocates into the nucleus, where it binds to specific DNA sequences known as glucocorticoid response elements (GREs), located at the regulatory regions of target genes. This interaction enhances the transcription and expression of anti-inflammatory genes while simultaneously suppressing the expression of pro-inflammatory mediators [48]. Nuclear factor kappa B (NF-κB) is a key transcriptional regulator involved in inflammatory responses. Its excessive activation within the nucleus leads to the production of various cytokines and chemokines, contributing to chronic inflammation and diseases such as severe asthma. The interaction between GR and NF-κB results in mutual inhibition; GR binding prevents NF-κB from associating with its target gene promoters, thereby suppressing the transcription of inflammation-related genes [50]. Similarly, activator protein 1 (AP-1), another critical regulator of inflammatory gene expression, is also inhibited through direct interaction with the GR, leading to the reduced synthesis of inflammatory molecules. In addition, the GR can modulate inflammatory responses by interacting with other transcription factors, including members of the signal transducer and activator of transcription family, further broadening its anti-inflammatory regulatory functions [51]. Beyond genomic effects, GCs also exhibit non-genomic actions, which differ significantly in terms of onset time and mechanism. These rapid effects typically manifest within seconds to minutes following hormone exposure [52]. Non-genomic mechanisms include the direct modulation of the MAPK signaling pathway and interactions with membrane-associated GRs. These pathways enable swift cellular responses that complement the slower genomic actions [53].

GR signaling plays a multifaceted role in cancer biology, with its functional outcomes being highly context-dependent. Preclinical studies have demonstrated that GR activity varies depending on the tumor type and cellular environment, acting either as a tumor suppressor or an oncogenic driver. Accumulating evidence indicates that GR signaling participates in the regulation of multiple biological processes in both hematological malignancies and solid tumors, including lung cancer, prostate cancer, breast cancer, pancreatic cancer, and bladder cancer [54]. In the context of NSCLC, substantial experimental data support a tumor-suppressive role for the GR (Figure 2). Greenberg et al. [55] reported that dexamethasone, a synthetic glucocorticoid, induces cell cycle arrest in A549 lung adenocarcinoma cells via GR activation. This effect is associated with the decreased phosphorylation of retinoblastoma protein (Rb), reduced activity of cyclin-dependent kinase 2 (CDK2) and CDK4, and the downregulation of cyclin D/E2, E2F transcription factor, and Myc protein expression, along with increased levels of the CDK inhibitor p21. Furthermore, dexamethasone exerts its anti-proliferative effects primarily through the GR-mediated inhibition of the ERK and MAPK signaling pathways. Srivastava et al. [37] further demonstrated that dexamethasone suppresses the migratory, invasive, and colony-forming capacities of A549 cells by disrupting the organization of cytoplasmic actin. Concurrently, it induces cell cycle arrest through the upregulation of CDK inhibitors p21 and p27, along with the downregulation of CDK4/6 expression. Collectively, these findings illustrate that the tumor-suppressive function of GR signaling in NSCLC is well supported by diverse molecular mechanisms, including cell cycle regulation, the suppression of oncogenic signaling pathways, and the induction of cellular senescence. The role of GR in cancer remains highly context-dependent, but, in NSCLC, it clearly contributes to growth inhibition and represents a potentially valuable therapeutic target.

Nonetheless, accumulating evidence suggests that GR signaling may exert oncogenic effects under specific contexts. A recent study by Lakshmanan et al. demonstrated that the MUC16-mediated phosphorylation of JAK2 and STAT3 facilitates the nuclear translocation of STAT3 and its subsequent interaction with the GR. The resultant STAT3-GR complex binds to the GRE-containing promoter region of the testis-specific Y-like protein 5 (TSPYL5) gene, thereby upregulating its expression and promoting the proliferation and migration of NSCLC cells [56]. Another study further supporting the oncogenic potential of GR signaling revealed that GR activation can drive cancer cells into a reversible dormant state, characterized by resistance to multiple chemotherapeutic agents and increased sensitivity to inhibitors of the insulin-like growth factor 1 receptor (IGF1R)-mediated survival pathway [16]. Similarly, Yang et al. [57] reported that the stress-induced elevation of plasma corticosterone levels correlates with the upregulation of TSC22 domain family protein 3 (TSC22D3), a glucocorticoid-induced transcriptional regulator with immunosuppressive properties. This factor has been shown to impair the type I interferon response in dendritic cells and suppress T-cell activation.

Notably, steroid hormone signaling does not act in isolation but rather is integrated with several key oncogenic pathways in NSCLC, such as EGFR, PI3K/AKT, MAPK, and STAT3. These interactions contribute to diverse functional outcomes, including cell proliferation, survival, immune evasion, and therapy resistance. Understanding this crosstalk is critical in identifying synergistic therapeutic strategies and predicting treatment responses. Table 2 summarizes the key interactions between steroid hormone receptors and major oncogenic pathways in NSCLC (Table 2).

## 4. Clinical Application of Steroid Hormones in the Treatment of NSCLC

Steroid hormones and their receptors have emerged as potential modulators of tumor progression and therapeutic responses in NSCLC. Building on insights from hormone-driven cancers such as breast and prostate cancer, recent research has explored the clinical relevance of estrogen, progesterone, androgen, and glucocorticoid pathways in NSCLC. These hormones not only influence tumor cell proliferation and survival but also affect treatment resistance, immune regulation, and the tumor microenvironment. This section summarizes current clinical evidence regarding the use of hormone modulators—either alone or in combination with chemotherapy, targeted therapy, or immunotherapy—and discusses ongoing challenges and therapeutic prospects in hormone-based interventions for NSCLC.

### 4.1. Application of GCs in the Treatment of NSCLC

Recent findings by Parajuli et al. [58] have identified a novel strategy to improve the therapeutic efficacy of immunotherapy in NSCLC. Their study demonstrated that the pretreatment of lung adenocarcinoma cells with dexamethasone can induce cellular senescence, thereby promoting the increased infiltration of T cells and natural killer (NK) cells into the tumor microenvironment. This immune-enhancing effect may contribute to improved responses to immunotherapeutic interventions in NSCLC patients. Postoperative complications following lung cancer resection remain a significant clinical concern. Among these, pneumonia is one of the most frequently observed, with reported incidence rates ranging from 10% to 30%. Of these cases, approximately 5% are classified as severe pneumonia, associated with a mortality rate of up to 9% [59]. In such scenarios, GCs are often administered as an adjunctive therapy alongside antibiotics. These agents help to reduce inflammatory exudation in lung tissues, enhance systemic tolerance to endotoxins, and restore cortisol levels in patients experiencing relative adrenal insufficiency due to infection [60]. Another common complication is intraoperative or postoperative bleeding, particularly during early-stage lung cancer surgery, where the need for blood transfusion occurs in 1% to 10% of cases. Allergic reactions to blood transfusions are not uncommon and can lead to significant morbidity if not properly managed. GCs have been shown to suppress the release of allergic mediators and are routinely used as prophylactic agents prior to transfusion to mitigate the risk of hypersensitivity reactions [61,62].

In summary, GCs play a multifaceted role in the clinical management of NSCLC, spanning from enhancing anti-tumor immunity to managing postoperative complications. Their immunomodulatory and anti-inflammatory properties make them valuable components of both surgical and oncologic treatment protocols.

### 4.2. Potential of Anti-Estrogen Therapy

Estrogen, progesterone, and their respective receptors have been increasingly recognized for their roles in the initiation, progression, and clinical outcomes of NSCLC. As a result, anti-hormonal therapy has emerged as a promising therapeutic strategy for selected NSCLC patients. Some researchers propose that disrupting estrogen signaling—through mechanisms such as inhibiting its biosynthesis, promoting its metabolic degradation, or blocking its interaction with the ER—may represent a novel approach in lung cancer treatment. Anti-estrogen agents are primarily categorized into three classes: selective estrogen receptor modulators (SERMs), pure anti-estrogens, and aromatase inhibitors (AIs) [13]. The ER has been explored as a potential therapeutic target in lung cancer. Preclinical studies have demonstrated that fulvestrant, a potent pure anti-estrogen, significantly suppresses the growth of subcutaneous xenograft tumors in ovariectomized nude mice. Notably, this inhibitory effect is further enhanced when fulvestrant is combined with erlotinib, an EGFR-TKI, whereas erlotinib alone shows limited efficacy in reducing the tumor volume in these models [63]. In vitro experiments have similarly confirmed a synergistic anti-tumor effect when anti-estrogens are used in combination with TKIs [64]. Furthermore, combination therapy has been shown to reduce the basal secretion of VEGF by lung cancer cells. This mechanism may involve the suppression of ERβ-mediated VEGF production, thereby impairing tumor angiogenesis. The observed synergy between anti-estrogens and EGFR inhibitors is likely due to the crosstalk between estrogen signaling and growth factor pathways. Supporting this hypothesis, the combination of fulvestrant and vandetanib—a multi-targeted tyrosine kinase inhibitor—has been found to significantly inhibit tumor growth in xenografted mice compared to either agent administered as a monotherapy [20].

Multiple preclinical datasets also indicate that fulvestrant exerts anti-proliferative effects on various NSCLC cell lines. Synergistic interactions have been consistently observed when anti-estrogens are co-administered with EGFR inhibitors, reinforcing the rationale for dual targeting strategies in hormone-responsive lung cancers [65]. Importantly, anti-estrogen therapy is no longer confined to preclinical investigations. Clinical evidence has begun to emerge, including a phase II trial evaluating the combination of fulvestrant and gefitinib in 22 postmenopausal women with advanced NSCLC. The results revealed that patients with ERβ expression levels exceeding 60% exhibited a median overall survival duration of 65.5 weeks, compared to only 21 weeks in those with ERβ expression below 60% [66]. A study has shown that the combination of the ER antagonist fulvestrant and the TLR4-specific inhibitor CLI-095 exhibits a synergistic anti-metastasis effect on human NSCLC cells. Through the immunohistochemical analysis of 180 primary NSCLC samples and 30 corresponding metastatic lymph node samples, it was found that the expression of ERβ and TLR4 was positively correlated in both primary NSCLC tissues and metastatic lymph node tissue. Experimental results demonstrated that either fulvestrant or CLI-095 alone could inhibit the migration and invasion of NSCLC cells, while the combined use of the two showed the strongest inhibitory effect. Moreover, CLI-095 could also help fulvestrant to limit the formation and function of invasive pseudopodia in NSCLC cells [67]. These findings suggest that ERβ status may serve as a predictive biomarker for the response to anti-estrogen-based therapies. As highlighted by Stabile et al. [68], estrogen can promote the proliferation and tumor growth of NSCLC cells. Targeting estrogen signaling through anti-estrogen interventions has been shown to reduce the tumor burden, inhibit cancer cell proliferation, and potentially improve patient outcomes. These data collectively support the exploration of anti-estrogen therapy as a viable and biologically rational approach in the management of NSCLC, particularly in molecularly defined subpopulations [20].

### 4.3. Combination Therapy with Steroid Hormones and Immune Checkpoint Inhibitors

Tumor immunotherapy has emerged as the fifth major therapeutic modality in lung cancer treatment, following surgery, radiotherapy, chemotherapy, and targeted therapy. It represents one of the most promising and effective strategies for the management of NSCLC. The combination of anti-estrogen therapy with immunotherapy offers additional advantages, as steroid hormones are known to exert regulatory effects on immune system function and may influence the efficacy of immune-based interventions. Steroid hormones, including estrogen and progesterone, play a crucial role in modulating both innate and adaptive immune responses. These hormonal influences can affect the tumor microenvironment composition, immune cell infiltration, and the expression of immune checkpoint molecules. Therefore, integrating anti-hormonal strategies with immunotherapy may enhance anti-tumor immunity and improve clinical outcomes in NSCLC patients [69]. In recent years, numerous clinical trials have established the efficacy of immune checkpoint inhibitors (ICIs) in the treatment of NSCLC. ICIs demonstrate significant clinical benefits, not only as a monotherapy but also when combined with chemotherapy. Currently, multiple ongoing studies are investigating their application in neoadjuvant and adjuvant settings, aiming to improve long-term survival and reduce recurrence rates [70].

The most commonly used ICIs target programmed death receptor 1 (PD-1), programmed death ligand 1 (PD-L1), and cytotoxic T-lymphocyte-associated antigen 4 (CTLA-4) [71,72]. Compared to conventional chemotherapy, these agents offer higher response rates, greater specificity, and more durable clinical benefits [73]. Notably, Patel et al. [74] reported that estrogen may contribute to resistance to bevacizumab—an anti-angiogenic agent—by promoting tumor vascular maturation and recruiting immunosuppressive myeloid cells. This finding suggests that blocking estrogen signaling through fulvestrant could potentially enhance the effectiveness of anti-angiogenic therapies. Moreover, it provides a mechanistic basis for understanding sex-related differences in the treatment response and supports the rationale for combining hormone-modulating agents with immunotherapeutic approaches in NSCLC management.

However, when it comes to another class of hormones—GCs—their impact on ICI efficacy appears to be predominantly negative. Studies have indicated that the administration of high-dose GCs may negatively impact the therapeutic efficacy of ICIs [75]. The proposed mechanisms involve the suppression of the activation, proliferation, and functional activity of key immune cells, including T cells, NK cells, and dendritic cells, thereby diminishing the immune system’s capacity to recognize and eliminate tumor cells [76]. Moreover, high-dose GCs have been shown to enhance the expression of PD-L1 on tumor cell surfaces. This molecule interacts with PD-1 to inhibit T-cell responses, thereby promoting tumor cell resistance to ICI therapy [77].

## 5. Future Directions and Challenges

Despite significant advances in understanding the roles of steroid hormones in NSCLC, several key challenges remain that warrant focused investigation. Future research directions should aim to address these limitations and explore new therapeutic opportunities.

### 5.1. Elucidation of Hormone Signaling Crosstalk and Resistance Mechanisms

Converging data indicate that estrogen signaling intersects with EGFR, MAPK/ERK, PI3K/AKT, and angiogenic programs in NSCLC. ERβ-selective activation increases p44/42-MAPK and proliferation in lung cancer cells [46]. Estrogen–growth factor crosstalk enhances EGFR phosphorylation and VEGF output, and preclinical studies show synergies when combining anti-estrogens with EGFR inhibitors (fulvestrant with erlotinib/vandetanib), including reduced VEGF secretion and tumor growth suppression [41]. A pilot clinical study further suggests that tumor patients with ERβ >60% derive longer overall survival from gefitinib plus fulvestrant than those with lower ERβ expression (65.5 vs. 21 weeks), highlighting ERβ as a predictive marker [66]. Estrogen can also diminish the anti-angiogenic efficacy in murine NSCLC, implying a rationale for the addition of ER blockade to bevacizumab-based regimens in selected patients [74].

On the glucocorticoid axis, GR activation suppresses ERK signaling and induces G1 arrest/senescence in NSCLC models [55], but the GR also cooperates with activated STAT3 to transactivate TSPYL5, fostering growth and chemoresistance [78]. In LKB1-mutant NSCLC, dexamethasone induces CDKN1C/p57 and G1/S arrest, highlighting a genotype-conditioned vulnerability [39]. Clinically, baseline or concomitant GCs may blunt anti-tumor immunity and impact ICI outcomes, underscoring schedule- and dose-dependent trade-offs [10]. Non-genomic escape via GPER further sustains MAPK/PKA signaling and can be antagonized by G15 [42].

These data support testable co-targeting strategies (ER/EGFR, ER/angiogenesis, context-specific GR/STAT3) and biomarker-driven designs using ERβ IHC thresholds, LKB1/CPS1 status, and GPER expression. To dissect compensatory networks and nominate synthetic–lethal nodes, future work should integrate phosphoproteomics and single-cell multi-omics with CRISPR perturbation screens anchored to these biomarkers.

### 5.2. Development of Selective Hormone-Based Therapeutics

Conventional GCs and anti-estrogens lack tumor specificity and can induce systemic adverse effects, including immune suppression, which may attenuate responses to ICIs [10]. To improve the therapeutic precision, several selective strategies within the estrogen and glucocorticoid axes merit prioritization. On the estrogen pathway, preclinical studies demonstrate that combining fulvestrant with EGFR inhibitors (erlotinib/vandetanib) yields synergistic growth inhibition and reduces VEGF output, providing a mechanistic rationale for dual targeting [41]. A pilot clinical study further suggests that tumor patients with ERβ >60% experience longer overall survival with gefitinib + fulvestrant than those with ERβ-low tumors (65.5 vs. 21 weeks), highlighting ERβ as a predictive biomarker for endocrine-targeted combinations [66]. Beyond nuclear ERs, GPER drives rapid, non-genomic signaling; the GPER antagonist G15 reverses E2-induced proliferation in NSCLC models, supporting GPER as a selective drug target [42].

Within the GC pathway, mechanistic insights indicate opportunities for biased/selective GR modulation. The GR’s anti-inflammatory transrepression can be separated from metabolic transactivation, a principle that underpins the development of selective GR modulators [53]. HDAC2-dependent deacetylation enhances the GR-mediated repression of NF-κB, highlighting an epigenetic lever to augment the anti-inflammatory efficacy while potentially minimizing off-target effects [50]. At the same time, the genotype can condition the GC benefit: in LKB1-mutant NSCLC, dexamethasone induces CDKN1C/p57 and G1/S arrest, nominating LKB1/CPS1 as stratification markers for GR-directed interventions. Given the ICI context, the dose and scheduling of GR-active agents should be prospectively optimized to preserve anti-tumor immunity [11].

Collectively, these data support a development path centered on selectivity plus biomarker enrichment—e.g., ERβ-high/GPER-positive tumors for endocrine combinations, and LKB1/CPS1-defined subsets for GR-modulatory strategies—accompanied by pharmacodynamic readouts (pERK suppression, p21/p57 induction) to confirm on-target activity.

### 5.3. Identification and Validation of Predictive Biomarkers

The clinical deployment of hormone-targeted strategies in NSCLC will depend on validated, assayable biomarkers. Among estrogen axis markers, nuclear ERβ shows prognostic and biological relevance and is associated with EGFR status in resected cohorts [23]. Importantly, a pilot trial reported longer overall survival for gefitinib + fulvestrant when ERβ IHC >60% versus ≤60% (65.5 vs. 21 weeks), supporting ERβ as a predictive threshold for endocrine–EGFR combinations [66]. Beyond nuclear ERs, GPER is detectable in NSCLC and mediates rapid signaling; its expression and druggability nominate it as a candidate biomarker for non-genomic estrogen dependence [42].

On the glucocorticoid axis, higher GR expression correlates with improved survival in advanced NSCLC, suggesting stratification value. Moreover, LKB1 mutation with CPS1 upregulation represents a context in which dexamethasone induces CDKN1C/p57 and G1/S arrest, indicating a genotype-conditioned vulnerability to GR-directed approaches [39]. For progesterone signaling, PR expression has been linked to outcomes in subsets and merits standardized evaluation alongside ERβ; ER/PR profiling also correlates with EGFR mutation in surgical series.

To enable routine use, we recommend harmonized IHC (antibodies, nuclear vs. cytoplasmic scoring, H-score cutoffs), prespecified thresholds (ERβ > 60%), and prospective, biomarker-enriched trials that embed pharmacodynamic readouts and genotype (LKB1/CPS1) to confirm on-target effects [39].

### 5.4. Addressing Sex- and Gender-Based Differences

Accumulating evidence suggests that the expression and functional consequences of steroid hormone receptors in NSCLC may differ between males and females. These differences can influence disease progression, prognosis, and treatment responsiveness. In addition to biological sex differences, gender—as a sociocultural construct encompassing identity, roles, and behaviors—may also influence lung cancer outcomes. For instance, gender-associated factors such as smoking behavior, occupational exposures, perceived health risks, and disparities in healthcare access may modulate the cancer risk and treatment responsiveness. Emerging evidence suggests that gender-diverse populations may face diagnostic delays and suboptimal treatment due to systemic bias or reduced access to gender-affirming care. While current data on the interaction between gender identity and steroid hormone signaling in NSCLC remain scarce, future studies should address this gap to ensure equitable and personalized cancer care across all gender identities.

Substantial gender disparities exist in NSCLC incidence, progression, and treatment responses, potentially due to differences in hormonal milieu and receptor expression. Future studies should stratify patients by sex and hormonal status to elucidate the impacts of sex steroids on tumor behavior and treatment outcomes. Personalized medicine approaches that account for gender-specific biological characteristics may improve efficacy and reduce adverse effects.

### 5.5. Managing the Dual Roles of GCs

GCs display context-dependent biology in NSCLC. Via the GR, GCs suppress ERK signaling and downregulate cyclin D/E2F/MYC while inducing p21/p27, leading to G1 arrest and senescence in lung cancer models [37]. Clinically, higher GR expression has correlated with improved survival, supporting stratification by GR status [38]. The genotype further conditions the benefit: in LKB1-mutant NSCLC, dexamethasone upregulates CDKN1C/p57 and enforces G1/S arrest, nominating LKB1/CPS1 as selection markers for GR-directed strategies [39]. Conversely, systemic GCs dampen T-cell activation and may attenuate responses to ICIs, necessitating careful dose and timing—particularly at ICI initiation [11]. Notably, prolonged dexamethasone exposure can induce tumor cell senescence with a SASP that recruits T/NK cells, suggesting schedule-dependent opportunities for GC-ICI integration [58]. Management should minimize baseline steroids when starting ICIs, explore selective/biased GR modulators, and embed pharmacodynamic readouts (pERK suppression, p21/p57 induction) in biomarker-enriched trials guided by GR, LKB1, and CPS1.

### 5.6. Enhancing Integration with Immunotherapy

ICIs have reshaped NSCLC care (nivolumab, pembrolizumab, and atezolizumab) [72]. Mounting evidence links steroid hormone pathways to immune tone, suggesting rational combinations. Estrogen signaling can promote angiogenesis and recruit immunosuppressive myeloid cells, potentially undermining anti-tumor immunity; the preclinical blockade of estrogen pathways mitigates these effects and may complement ICI activity [74]. Rapid, non-genomic estrogen signaling via GPER is druggable and represents an additional lever to recondition the tumor microenvironment [42]. On the glucocorticoid axis, dexamethasone-induced tumor cell senescence has been shown to increase T/NK-cell infiltration, implying schedule-dependent opportunities for GC-ICI integration [58]; however, baseline or early systemic steroids can blunt ICI efficacy, mandating careful dosing and timing [11]. We propose biomarker-driven trials stratified by ERβ/GPER and GR/LKB1/CPS1 status, with prespecified sequencing (endocrine lead-in, minimal steroids near ICI initiation) and pharmacodynamic readouts to confirm on-target immune modulation.

In conclusion, while steroid hormones represent a promising frontier in NSCLC biology and therapy, numerous scientific and clinical challenges must be addressed. Tackling these issues will require interdisciplinary collaboration, innovative methodologies, and well-designed clinical trials to translate mechanistic insights into effective and individualized treatment strategies.

## 6. Conclusions

Steroid hormones exert significant regulatory effects on NSCLC through both genomic and non-genomic pathways, mediated by sex hormones and their receptors. These hormonal signaling mechanisms influence key tumor biological processes, including cell proliferation, angiogenesis, invasion, and metastasis. Progesterone has been shown to inhibit tumor growth via the PR, while AR expression has been associated with the response to EGFR-TKI therapy. GCs mediate a range of biological effects through the GR, involving both genomic and rapid non-genomic mechanisms. These include the induction of cell cycle arrest, suppression of inflammatory cytokines, and inhibition of tumor cell migration and invasion. The clinical potential of steroid hormone-related pathways lies in several areas: the development and application of anti-estrogen therapies, the precise use of GCs in treatment protocols, and the utilization of hormone receptors as biomarkers for diagnosis, prognosis, and therapeutic stratification. Future research should focus on elucidating the crosstalk between different signaling pathways, understanding mechanisms of drug resistance, and facilitating the discovery of novel targeted therapeutics. However, several challenges remain to be addressed, including the immunosuppressive effects of GCs, heterogeneity in receptor expression across tumor subtypes, and the influence of gender-based differences on treatment outcomes. Overcoming these obstacles will be critical to optimize hormone-targeted strategies in the management of NSCLC.

## Figures and Tables

**Figure 1 biomedicines-13-01992-f001:**
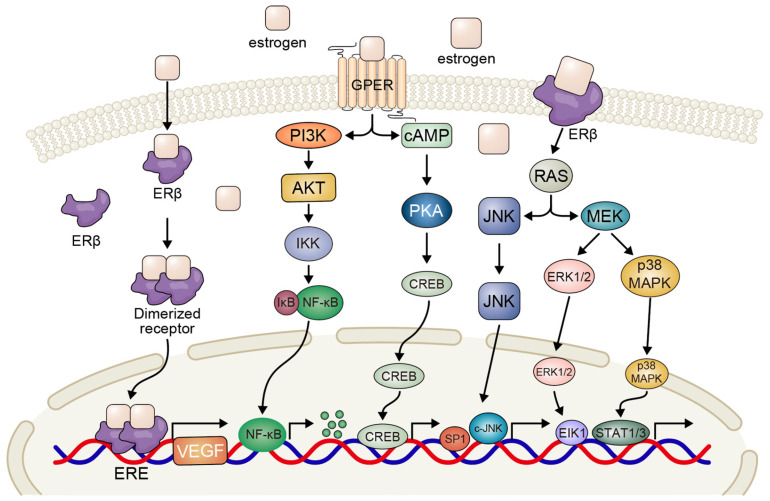
Schematic of estrogen-mediated signaling pathways in NSCLC. Estrogen exerts biological effects via ERβ-centered cascades. In the genomic pathway, the estrogen-bound ERβ dimer translocates to the nucleus, interacting with ERE or transcription factors to regulate the transcription of genes like Cyclin D1, VEGF, and Bcl-2. In the non-genomic pathway, ERβ activates cytoplasmic kinase cascades: PI3K/AKT promotes cell survival; MAPK and cAMP/PKA activate CREB for growth-related gene expression. GPER-mediated signaling triggers rapid RAS/MEK/ERK and PI3K/AKT activation for proliferation. NF-κB is linked to inflammation. (NSCLC: non-small cell lung cancer; ER: estrogen receptor; ERE: estrogen response element; VEGF: vascular endothelial growth factor; PI3K: phosphoinositide 3-kinase; AKT: protein kinase B; JNK: c-Jun N-terminal kinase; MAPK: mitogen-activated protein kinase; cAMP: cyclic adenosine monophosphate; MEK: mitogen-activated protein kinase; ERK: extracellular signal-regulated kinase; NF-κB: nuclear factor kappa B).

**Figure 2 biomedicines-13-01992-f002:**
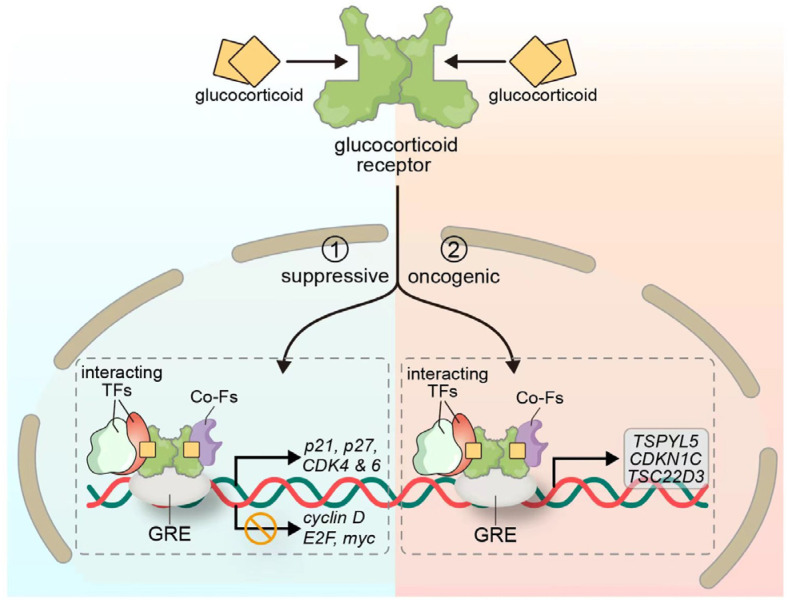
Schematic of glucocorticoid-related signaling impacts on NSCLC immunotherapy and tumor biology. In the cytoplasm, GR interaction suppresses the activation, proliferation, and function of key immune cells (T cells, NK cells, dendritic cells) via inhibiting cytokine production and metabolic pathways, reducing the anti-tumor immune capacity. In the tumor cell context, GCs enhance PD-L1 expression on tumor surfaces. PD-L1 then binds PD-1 on T cells, blocking T-cell responses. (NSCLC: non-small cell lung cancer; GR: glucocorticoid receptor; NK: natural killer; GCs: glucocorticoids; PD-L1: programmed death ligand 1; PD-1: programmed death receptor 1).

**Table 1 biomedicines-13-01992-t001:** Steroid hormone receptors in NSCLC—expression and clinical implications.

Receptor	Expression in NSCLC	Gender Association	Prognostic Value	Therapeutic Implications	Reference
ERα/ERβ	ERβ ↑ (more common)	↑ in females	Favorable (ERβ)	Anti-estrogens	[12,13]
PR	Low/moderate	Variable	Unclear	Limited studies	[14]
AR	Low/moderate	↑ in males	Poor prognosis	Under investigation	[15]
GR	Widely expressed	No clear link	Context-dependent	Corticosteroids, GR modulators	[16]

Abbreviations: ER: estrogen receptor; PR: progesterone receptor; AR: androgen receptor; GR: glucocorticoid receptor.

**Table 2 biomedicines-13-01992-t002:** Crosstalk between steroid hormone signaling and oncogenic pathways in NSCLC.

Steroid Receptor	Crosstalk Pathway	Molecular Mechanism	Functional Outcomes	Therapeutic Implications	References
ERβ (±ERα)	EGFR	Estrogen promotes EGFR phosphorylation and activates downstream PI3K/AKT and MAPK signaling; reciprocal compensation between pathways	Enhances proliferation and survival; may lead to TKI resistance	Combining anti-estrogens with EGFR-TKIs to overcome resistance	[45]
ERβ	MAPK/ERK, PI3K/AKT	ERβ-selective ligands activate MAPK and AKT pathways	Promotes tumor cell growth and proliferation	Dual targeting of ERβ and MAPK/PI3K pathways	[43,44]
ER	VEGF	Estrogen increases VEGF expression; anti-estrogen + EGFR-TKI reduces VEGF secretion	Promotes angiogenesis and tumor vascularization	Use of anti-estrogens in combination with anti-angiogenic or EGFR-targeted therapies	[40,41]
GPER	MAPK, cAMP/PKA	Membrane-bound GPER activates non-genomic cascades; G15 inhibits E2-induced proliferation	Stimulates rapid proliferation and migration	GPER antagonists as novel anti-proliferative agents	[42]
GR	STAT3	GR cooperates with activated STAT3 to transactivate TSPYL5, which suppresses p53	Promotes tumor growth and drug resistance	Targeting STAT3-GR axis or TSPYL5 expression to restore drug sensitivity	[56]
GR	NF-κB/AP-1	GR inhibits NF-κB and AP-1 via protein–protein interaction; HDAC2-mediated deacetylation enhances repression	Reduces inflammation but may suppress anti-tumor immunity	Use selective GR modulators to balance anti-inflammatory and immune activation	[50,51]
GR	ERK/MAPK and Cell Cycle	Dexamethasone via GR suppresses ERK signaling, downregulates Cyclin D/E2F/Myc, upregulates p21/p27	Induces cell cycle arrest and senescence	GR agonism beneficial in LKB1-mutant NSCLC or steroid-sensitive tumors	[37]

Abbreviations: ER: estrogen receptor; GPER: G-protein-coupled estrogen receptor; GR: glucocorticoid receptor.

## Data Availability

No new data were created or analyzed in this study.

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
