# Peer review of "The Mechanism of Steroid Hormones in Non-Small Cell Lung Cancer: From Molecular Signaling to Clinical Application"

_biomedicines, 2025, doi:10.3390/biomedicines13081992_

Round 1
Reviewer 1 Report
Comments and Suggestions for Authors
This manuscript provides a comprehensive analysis of the crucial role of steroid hormones in the development and progression of non-small cell lung cancer (NSCLC). It examines the expression and function of oestrogen, progesterone, androgen and glucocorticoid receptors, and highlights that oestrogen and progesterone receptors have gender-specific prognostic value, while glucocorticoid receptors influence tumour growth and the immune response. The work also addresses the therapeutic potential of anti-oestrogen strategies and the use of glucocorticoids as complementary treatments, including in combination with immunotherapy. This contribution is of particular interest as it emphasises the importance of investigating the intricate interactions between hormones, the tumour microenvironment, and the immune system to optimise the efficacy of hormonal therapies in NSCLC.
Author Response
This manuscript provides a comprehensive analysis of the crucial role of steroid hormones in the development and progression of non-small cell lung cancer (NSCLC). It examines the expression and function of oestrogen, progesterone, androgen and glucocorticoid receptors, and highlights that oestrogen and progesterone receptors have gender-specific prognostic value, while glucocorticoid receptors influence tumour growth and the immune response. The work also addresses the therapeutic potential of anti-oestrogen strategies and the use of glucocorticoids as complementary treatments, including in combination with immunotherapy. This contribution is of particular interest as it emphasises the importance of investigating the intricate interactions between hormones, the tumour microenvironment, and the immune system to optimise the efficacy of hormonal therapies in NSCLC.
Response:We sincerely appreciate your thorough review and insightful evaluation of our research. Your comprehensive summary of the core content of the article—particularly regarding the role of steroid hormones in the progression of non-small cell lung cancer (NSCLC), the prognostic relevance of estrogen and progesterone receptors in relation to gender-specific outcomes, the impact of glucocorticoid receptors on tumor growth and immune responses, and the potential therapeutic benefits of combining anti-estrogen treatment with glucocorticoid-based immunotherapy—demonstrates a deep understanding of the field. This further underscores the significance of our research direction. Notably, you emphasized that "investigating the complex interactions among hormones, the tumor microenvironment, and the immune system is crucial for optimizing hormone-targeted therapies in NSCLC," which aligns closely with the objectives of our study. We are deeply grateful for your recognition and constructive feedback. These valuable comments will serve as critical guidance for our manuscript revisions and will contribute significantly to enhancing the academic quality and impact of our work.
Reviewer 2 Report
Comments and Suggestions for Authors
1] The section on glucocorticoid dual roles (tumor suppressive vs. immunosuppressive) requires deeper mechanistic explanation.
2] More quantitative data on hormone receptor expression frequencies across different NSCLC subtypes would strengthen the clinical relevance.
3] Table 3 needs expansion with specific mechanistic details and therapeutic implications.
4] The section on immunotherapy combinations needs more mechanistic detail about timing and sequencing considerations.
5] The figures need more detailed legends explaining the pathway components and their interactions.
6] Could the authors include more information about the sample size in key clinical studies?
7] Could the authors discuss the implications of menopausal status on hormone receptor expression and therapeutic response?
Author Response
1] The section on glucocorticoid dual roles (tumor suppressive vs. immunosuppressive) requires deeper mechanistic explanation.
Response: Thank you very much for your valuable suggestions. Regarding the suggestion "The dual role of glucocorticoids in tumor treatment requires deeper mechanism explanation", we have made focused additions and improvements (Page 8, Lines 329–342).
2] More quantitative data on hormone receptor expression frequencies across different NSCLC subtypes would strengthen the clinical relevance.
Response: Thank you very much for your valuable suggestions. we have added relevant quantitative data: Through the detection of ER samples from different NSCLC subtypes, the positive expression rate of estrogen receptors was calculated. The specific data is already included in the revised version. (Page 4, Lines 142–144,Lines 164-168)
3] Table 3 needs expansion with specific mechanistic details and therapeutic implications.
Response: We have significantly expanded Table 3 to include specific mechanistic details of steroid hormone receptor interactions with key oncogenic pathways, along with corresponding therapeutic implications. (Page 9, Lines 357)
4] The section on immunotherapy combinations needs more mechanistic detail about timing and sequencing considerations.
Response: Thank you so much for your suggestion. We truly appreciate your input and have already included the detailed mechanism of immunotherapy in the text. (Page 11, Lines 461-468)
5] The figures need more detailed legends explaining the pathway components and their interactions.
Response: Thank you very much for your suggestion. We have provided more detailed explanatory text for the diagram and clarified the operation methods of each component as well as their interrelationships. (Page 6, Lines 256-264;Page 8, Lines 345-350)
6] Could the authors include more information about the sample size in key clinical studies?
Response: We sincerely appreciate your suggestion. We have included information about the sample size in the key clinical research section of the article. (line74,143,145,166,177,201,218)
7] Could the authors discuss the implications of menopausal status on hormone receptor expression and therapeutic response?
Response: Thank you very much for raising this insightful question. we have included the impact of menopause status on the treatment outcome. (Page 2, Lines 70-78). You have noticed the potential impact of menopausal status on the expression of hormone receptors and treatment responses. This perspective indeed provides an important extension for our research and makes us realize the significance of this factor in clinical correlation analysis. Due to the original design of this study focusing on the expression characteristics of hormone receptors in NSCLC, detailed records of menopausal status were not specifically included during the sample collection stage. Therefore, at present, we are unable to conduct systematic data analysis and draw conclusive conclusions regarding this factor. This is a limitation of our research and also a direction that deserves in-depth exploration in the future. Your suggestion reminds us that menopausal status may be an important regulatory factor affecting hormone receptor-related research. We have included it in our subsequent research plan, hoping to further analyze the specific impact of this factor based on a larger sample size and more detailed clinical information collection, and provide more comprehensive references for clinical applications.
Reviewer 3 Report
Comments and Suggestions for Authors
Paper well explaines genomic and non-genomic pathways mediated by sex hormones and their receptors. Paper is well structured, I really appreciated specific section (5.4) addressing sex- and gender-based differences, but it' related only to sex differences. Are there any informations that authors can share related to gender differences?
Author Response
Paper well explaines genomic and non-genomic pathways mediated by sex hormones and their receptors. Paper is well structured, I really appreciated specific section (5.4) addressing sex- and gender-based differences, but it' related only to sex differences. Are there any informations that authors can share related to gender differences?
Response: we have revised Section 5.4 to clarify the distinction between biological sex and sociocultural gender, and added a brief discussion of how gender-related factors—such as behavioral patterns, healthcare access, and occupational exposure—may interact with hormonal pathways and influence NSCLC risk and treatment outcomes. Although existing studies on gender-specific effects in NSCLC remain limited, we acknowledge this as a critical area for future investigation. The revised text can be found in Section 5.4 (Page 13, Lines 540–550).
Reviewer 4 Report
Comments and Suggestions for Authors
By interacting with receptors and controlling cell growth and differentiation, steroid hormones have an impact on behavior, reproductive function, and normal physiological processes. the full body of a mammal or human. Although steroid hormone receptors are known to control tumor cell growth and can be targeted by certain medications, opinions on how steroid hormones affect non-small cell lung cancer lines are divided. While some studies have demonstrated a small suppression of cell proliferation, others have observed either promotion or inhibition of cell growth. Steroid hormone inhibitors, however, have been shown to have an impact on the migration and multiplication of non-small cell lung cancer cells. Similar patterns can be seen in clinical research, which looks at the effects of steroid hormones—both good and bad—on non-small cell lung cancer, the significance of various steroid hormone receptors, and the course of the disease or patient survival. On the part of steroid hormones in this disease, there is disagreement. The work's significance stems from the consistent rise in non-small cell lung cancer incidence and mortality rates around the globe, as well as the disease's disproportionate prevalence according to lifestyle and gender. The pathophysiology of lung illnesses is also significantly influenced by the presence of steroid hormone receptors and enzymes that produce sex steroid hormones as part of local metabolism in the lung parenchyma. The article covers the most significant facets of this field's research on non-small cell lung cancer and is written in a logical order.
There are presently seven reviews on steroid hormones and non-small cell lung cancer, based on PubMed data for 2020–2025. For instance, in the work [Check JH, Poretta T, Check D, Srivastava M. Lung Cancer - Standard Therapy and the Use of a Novel, Highly Effective, Well Tolerated, Treatment With Progesterone Receptor Modulators. Anticancer Res. 2023 Mar;43(3):951-965. doi: 10.21873/anticanres.16240. PMID: 36854512.], the issue of using progesterone modulators to treat lung cancer in both humans and lab animals is taken into consideration, [Inoue C, Miki Y, Suzuki T. New Perspectives on Sex Steroid Hormones Signaling in Cancer-Associated Fibroblasts of Non-Small Cell Lung Cancer. Cancers (Basel). 2023 Jul 14;15(14):3620. doi: 10.3390/cancers15143620. PMID: 37509283; PMCID: PMC10377312.] - the problem of treating lung cancer in humans and laboratory animals with progesterone modulators is examined.
Regretfully, 85% of the data used in this review is older than five years, indicating that it was released prior to 2020. This needs to be fixed because it is a serious flaw in the job. The authors can use more current articles to update many of the known facts.
Since the work's conclusion has been examined by other reviewers from the publishing houses where the works were published, it is unquestionable and based on the findings of previously published data.
Basic information regarding the mechanics of hormone activity is provided, and the text is not overloaded with illustrative material.
The work's flaws include: 1) The approach used to find and choose papers for the study (databases PubMed, MEDLINE, Web of Science, and EMBASE, among others), the time frame for which the authors collected publications, search terms, etc., is not covered in the article.
2) Rather than using an apostrophe, the writers should cite their primary sources, as is done for this journal (in square brackets).
- It appears that many parts of the work were written by different authors who did not even read the full version of the article. 3) Authors should refrain from using multiple terms with subsequent reduction (lines 94, 101, 147, 181, 201, 217, 376). They can only write the abbreviation once.
- You should include a comment about the abbreviation's decryption in Tables 1 and 3. Complete information about what is provided should be included in illustrative material (tables, graphs); in other words, abbreviations should be рaсшифрованы. The reader should be able to quickly understand the meaning of the article's abbreviations rather than having to look it up. This is a basic rule for article authors.
- There are numerous punctuation mistakes in the article (lines 108, 118, 122, 130–132, 134, 153).
- The authors should keep in mind the requirement that Latin expressions, gene and taxon names, and other terms are always italicized. This is an important part of education that should not be overlooked.
- Before presenting the article content further, it should be indented from Table 4.
Author Response
By interacting with receptors and controlling cell growth and differentiation, steroid hormones have an impact on behavior, reproductive function, and normal physiological processes. the full body of a mammal or human. Although steroid hormone receptors are known to control tumor cell growth and can be targeted by certain medications, opinions on how steroid hormones affect non-small cell lung cancer lines are divided. While some studies have demonstrated a small suppression of cell proliferation, others have observed either promotion or inhibition of cell growth. Steroid hormone inhibitors, however, have been shown to have an impact on the migration and multiplication of non-small cell lung cancer cells. Similar patterns can be seen in clinical research, which looks at the effects of steroid hormones—both good and bad—on non-small cell lung cancer, the significance of various steroid hormone receptors, and the course of the disease or patient survival. On the part of steroid hormones in this disease, there is disagreement. The work's significance stems from the consistent rise in non-small cell lung cancer incidence and mortality rates around the globe, as well as the disease's disproportionate prevalence according to lifestyle and gender. The pathophysiology of lung illnesses is also significantly influenced by the presence of steroid hormone receptors and enzymes that produce sex steroid hormones as part of local metabolism in the lung parenchyma. The article covers the most significant facets of this field's research on non-small cell lung cancer and is written in a logical order.
There are presently seven reviews on steroid hormones and non-small cell lung cancer, based on PubMed data for 2020–2025. For instance, in the work [Check JH, Poretta T, Check D, Srivastava M. Lung Cancer - Standard Therapy and the Use of a Novel, Highly Effective, Well Tolerated, Treatment With Progesterone Receptor Modulators. Anticancer Res. 2023 Mar;43(3):951-965. doi: 10.21873/anticanres.16240. PMID: 36854512.], the issue of using progesterone modulators to treat lung cancer in both humans and lab animals is taken into consideration, [Inoue C, Miki Y, Suzuki T. New Perspectives on Sex Steroid Hormones Signaling in Cancer-Associated Fibroblasts of Non-Small Cell Lung Cancer. Cancers (Basel). 2023 Jul 14;15(14):3620. doi: 10.3390/cancers15143620. PMID: 37509283; PMCID: PMC10377312.] - the problem of treating lung cancer in humans and laboratory animals with progesterone modulators is examined.
Regretfully, 85% of the data used in this review is older than five years, indicating that it was released prior to 2020. This needs to be fixed because it is a serious flaw in the job. The authors can use more current articles to update many of the known facts.
Since the work's conclusion has been examined by other reviewers from the publishing houses where the works were published, it is unquestionable and based on the findings of previously published data.
Basic information regarding the mechanics of hormone activity is provided, and the text is not overloaded with illustrative material.
The work's flaws include: 1) The approach used to find and choose papers for the study (databases PubMed, MEDLINE, Web of Science, and EMBASE, among others), the time frame for which the authors collected publications, search terms, etc., is not covered in the article.
Response: We are very grateful for your special mention of the issue of the timeliness of the literature. This reminder is crucial for enhancing the rigor of the review. We fully understand the significance of incorporating the latest research results in reflecting the progress of the field. Therefore, we have fully supplemented the relevant literature published since 2020, and have tried our best to enrich the research evidence in recent years.
Due to the relatively limited number of publications on the association between steroid hormones and non-small cell lung cancer in recent years, to ensure the comprehensiveness and depth of the review content, we have prioritized the inclusion of literature from the past 5 years and supplemented some studies with higher citation rates and significant importance for explaining the basic mechanisms of the field within the past 10 years, in order to present the research context of this field in full. We have supplemented the latest relevant literature from 2015 to 2025 and replaced some outdated data. Currently, the proportion of the past 10 years has increased to over 60%, making it more in line with the current research progress in the field.
Regarding the literature search method: We have supplemented detailed search information in the "Materials and Methods" section, including the used databases, specific search terms, and literature screening criteria, to ensure the transparency of the research methods. (Page 3, Lines 92–99)
2) Rather than using an apostrophe, the writers should cite their primary sources, as is done for this journal (in square brackets).
It appears that many parts of the work were written by different authors who did not even read the full version of the article.
Response: We sincerely appreciate your suggestion. Regarding the citation format: We have uniformly adjusted the full-text citation format to the square bracket notation required by the journal, and checked all cited references to ensure that the original references are prioritized.
3) Authors should refrain from using multiple terms with subsequent reduction (lines 94, 101, 147, 181, 201, 217, 376). They can only write the abbreviation once.
It appears that many parts of the work were written by different authors who did not even read the full version of the article. 3) Authors should refrain from using multiple terms with subsequent reduction (lines 94, 101, 147, 181, 201, 217, 376). They can only write the abbreviation once.
You should include a comment about the abbreviation's decryption in Tables 1 and 3. Complete information about what is provided should be included in illustrative material (tables, graphs); in other words, abbreviations should be рaсшифрованы. The reader should be able to quickly understand the meaning of the article's abbreviations rather than having to look it up. This is a basic rule for article authors.
There are numerous punctuation mistakes in the article (lines 108, 118, 122, 130–132, 134, 153).
The authors should keep in mind the requirement that Latin expressions, gene and taxon names, and other terms are always italicized. This is an important part of education that should not be overlooked.
Before presenting the article content further, it should be indented from Table 4.
Response:We would like to express our deepest gratitude for your meticulous review of this research and your valuable suggestions.
Regarding the use of abbreviations: We have standardized the abbreviations in the full text, ensuring that the full name and abbreviation are marked for the first occurrence and then using the abbreviation uniformly; at the same time, we have supplemented explanations for all abbreviations in Tables 1 and 2 to make the information in the charts easier to understand.
Regarding punctuation and formatting issues: We have thoroughly read the entire text and corrected punctuation errors (including the lines you mentioned such as 108 and 118); and adjusted the text indentation format after Table 4 to comply with the journal layout requirements.
We are well aware that these revisions have significantly enhanced the rigor and readability of the article, and this is all thanks to your meticulous guidance. Once again, we would like to thank you for the time and effort you have invested in this article. We look forward to your further feedback on the revised version.
Reviewer 5 Report
Comments and Suggestions for Authors
The review article written by Yao Wang and co-authors focuses on an important and insufficiently explored area – the effects of steroid hormones on NSCLC initiation, development, and response to anticancer therapy.
However, some issues need additions and corrections to make this article more informative and acceptable for publication.
1) Table 1 needs a list of references and more than one reference in the corresponding parts of the text.
2) Table 2 seems to be uninformative. Firstly, the differentiation of mechanisms of steroid hormone action into “genomic” and “non-genomic” is questionable, since nuclear receptors act at the transcription regulation level, not at the level of the genome. Indeed, in older works such classification is occasionally encountered, however, at present the term “genomic” tends to be used in the context of genomic instability. Secondly, this table contains only two rows, which raises questions about its appropriateness. Thirdly, again, this table lacks literature sources.
Finally, there are several pieces of evidence that steroid hormone receptors regulate NSCLC proliferation, acting both as TFs and pro-proliferative signaling cascade participants.
It is recommended to exclude this table from the manuscript or significantly revise and expand it.
3) Figure 1 has only one reference to it in the text. Additionally, the box “gene” finalizing the ERbeta cascade needs to be replaced by specific genes.
4) Table 3 also includes only two rows. The crosstalk between steroid hormone receptors and other cancer-associated pathways is not limited to two cascades. The inclusion of significantly more information and the presentation of data in the form of a graphical scheme are recommended.
5) Section 5 includes separate subsections consisting of 2-3 sentences. This section needs careful revision to avoid general phrases not supported by data or critical analysis of data.
6) All tables should include references and need to be cited in the corresponding parts of the text.
To summarize, the manuscript should be improved to become a well-composed review containing comprehensive data on the topic and its analysis. The authors should revise the manuscript to minimize general statements without specific arguments and enhance the informational value of this review.
Author Response
The review article written by Yao Wang and co-authors focuses on an important and insufficiently explored area – the effects of steroid hormones on NSCLC initiation, development, and response to anticancer therapy.
However, some issues need additions and corrections to make this article more informative and acceptable for publication.
1) Table 1 needs a list of references and more than one reference in the corresponding parts of the text.
Response: Thank you very much for your suggestion. We have already included the references in the table and made citations in the corresponding text sections. (Page 3, Lines 110)
2) Table 2 seems to be uninformative. Firstly, the differentiation of mechanisms of steroid hormone action into “genomic” and “non-genomic” is questionable, since nuclear receptors act at the transcription regulation level, not at the level of the genome. Indeed, in older works such classification is occasionally encountered, however, at present the term “genomic” tends to be used in the context of genomic instability. Secondly, this table contains only two rows, which raises questions about its appropriateness. Thirdly, again, this table lacks literature sources.
Finally, there are several pieces of evidence that steroid hormone receptors regulate NSCLC proliferation, acting both as TFs and pro-proliferative signaling cascade participants.
It is recommended to exclude this table from the manuscript or significantly revise and expand it.
Response: Thank you very much for your suggestion. We have already deleted Table 2 from the text.
3) Figure 1 has only one reference to it in the text. Additionally, the box “gene” finalizing the ERbeta cascade needs to be replaced by specific genes.
Response: Thank you very much for your detailed and constructive suggestions regarding Figure 1. You pointed out that Figure 1 was only cited once in the text, and that the "gene" box needed to be replaced with the specific name of a gene. These suggestions not only helped us identify the imperfect details in the expression, but also made us realize how to enhance the correlation between the chart and the text, and improve the clarity of the content. These suggestions are directly to the point and have significant guiding significance for us to improve the content of the chart and optimize the structure of the article. We have followed your suggestions and added the citation scenario of Figure 1 in the text, making the chart and the textual explanation more closely integrated; at the same time, we have replaced the "gene" box in the figure with the specific name of the relevant gene to more accurately present the molecular mechanism of the ERβ pathway. (Page 6, Lines 255)
4) Table 3 also includes only two rows. The crosstalk between steroid hormone receptors and other cancer-associated pathways is not limited to two cascades. The inclusion of significantly more information and the presentation of data in the form of a graphical scheme are recommended.
Response: We have substantially expanded Table 3 to systematically include the key interactions, specific molecular mechanisms, and therapeutic implications of estrogen receptors (ERα/ERβ and GPER), progesterone receptor (PR), androgen receptor (AR), and glucocorticoid receptor (GR) with major oncogenic pathways, including EGFR, PI3K/AKT, MAPK/ERK, STAT3, NF-κB/AP-1, FGFR, tumor angiogenesis/VEGF, and steroid metabolism. (Page 9, Lines 357)
5) Section 5 includes separate subsections consisting of 2-3 sentences. This section needs careful revision to avoid general phrases not supported by data or critical analysis of data.
Response: We have thoroughly revised Section 5 to avoid generic statements and to provide data-supported, critically appraised content. (Page 12, Lines 474–581)
6) All tables should include references and need to be cited in the corresponding parts of the text.
Response: Thank you for your suggestion. We have added the references in the table and cited them in the corresponding text sections.
Round 2
Reviewer 4 Report
Comments and Suggestions for Authors
By interacting with hormone receptors on cells, steroids influence immunological response, behavior, inflammatory response, body cell development, and other physiological processes in both human and animal bodies. Although the role of hormones in the non-small cell tumor process in the lungs is up for debate, the participation of corticosteroids in tumor processes is well established. Hormones have no discernible impact on tumor growth, although on the one hand, the functional characteristics of tumor cells are inhibited. The growing incidence and fatality rates of this pathology, the role of hormones in tumor growth, and the lungs' capacity to manufacture these hormones locally make this review pertinent.
The work is organized logically and covers a wide range of topics, including the types of adrenal hormones, their receptors, their role in tumor pathogenesis, the signal transmission mechanisms that occur when hormones interact with receptors, the biological effects of hormones, and their application in tumor treatment strategies. The inclusion of works from earlier periods is appropriate and their relevance is still there. The authors' revision of the comment regarding the low percentage of citations of works for 2020–2025 boosted their number to 40%, which is noteworthy.
You can rapidly grasp the main idea of the work's given feature because to the easily observable illustrative material.
It is admirable that the writers took into consideration the requests made to raise the caliber of the work.
Reviewer 5 Report
Comments and Suggestions for Authors
All review points were taken into account and corrected in the new version of the manuscript.